

# TARANIS XGRE and IDEE Detection Capability of Terrestrial Gamma-Ray Flashes and Associated Electron Beams

David Sarria[1], Francois Lebrun[1,2], Pierre-Louis Blelly[3,4], Remi Chipaux[5], Philippe Laurent[1,2], Jean-Andre Sauvaud[3,4], Pierre Devoto[3,4], Damien Pailot[1], Jean-Pierre Baronick[1], and Miles Lindsey-Clark[1]

[1]APC, AstroParticule et Cosmologie, Universite Paris Diderot, CNRS/IN2P3, CEA/DRF/IRFU, Observatoire de Paris, Sorbonne Paris Cite, 10 rue Alice Domont et Leonie Duquet, F-75205 Paris Cedex 13, France
[2]CEA/DRF/IRFU/Sap, Bat. 709, Orme des Merisiers, CEA-Saclay, F-91191 Gif-sur-Yvette Cedex, France.
[3]Universite de Toulouse, UPS-OMP, IRAP, Toulouse, France.
[4]CNRS, IRAP, 9 Av. colonel Roche, Toulouse, France.
[5]CEA/DRF/IRFU/SEDI, CEA-Saclay, F-91191 Gif-sur-Yvette Cedex, France.

*Correspondence to:* David Sarria (dsarria@apc.in2p3.fr)

**Abstract.**

With a launch expected in 2018, the TARANIS micro-satellite is dedicated to the study of transient phenomena observed in association with thunderstorms. On-board the spacecraft, XGRE and IDEE are two instruments dedicated to study Terrestrial Gamma-ray Flashes (TGFs) and associated electron beams (TEBs). XGRE can detect electrons (energy range: 1 MeV to 10

MeV) and X/gamma-rays (energy range: 20 keV to 10 MeV), with a very high counting capability (about 10 million counts per second), and the ability to discriminate one type of particle from the other. The IDEE instrument is focused on electrons in the 80 keV to 4 MeV energy range, with the ability to estimate their pitch angles.

Monte-Carlo simulations of the TARANIS instruments, using a preliminary model of the spacecraft, allow sensitive area estimates for both instruments. It leads to an averaged effective area of 425 cm$^2$ for XGRE to detect X/gamma rays from

TGFs, and the combination of XGRE and IDEE gives an average effective area of 255 cm$^2$ to detect electrons/positrons from TEBs. We then compare these performances to RHESSI, AGILE, and Fermi GBM, using performances extracted from literature for the TGF case, and with the help of Monte-Carlo simulations of their mass models for the TEB case.

Combining this data with with the help of the MC-PEPTITA Monte-Carlo simulations of TGF propagation in the atmosphere, we build a self-consistent model of the TGF and TEB detection rates of RHESSI, AGILE, and Fermi. It can then be used to

estimate that TARANIS should detect about 225 TGFs/year and 25 TEBs/year.

## 1 Introduction

Terrestrial Gamma Ray flashes (TGFs) are short ( $\sim 20\ \mu$s to $\sim 1$ ms) X and gamma ray emissions associated with lightning and mostly detected from space. Together with transient luminous events for the optical part (see Surkov and Hayakawa (2012) for a comprehensive review), they play an important role for understanding the coupling between magnetosphere-ionosphere-

atmosphere. A comprehensive review of TGFs and related studies, so called the High Energy Atmospheric Physics, is provided



by Dwyer et al. (2012). Detections of TGFs from space were first presented by Fishman et al. (1994), using data from the the Burst And Transient Source Experiment (BATSE) on-board the NASA's Compton Gamma-Ray Observatory. In the subsequent years, TGFs were also detected from space by other satellites : the Reuven Ramaty High Energy Solar Spectroscopic Imager (RHESSI) (Smith et al., 2005), the Astro-rivelatore Gamma a Immagini Leggero (AGILE) (Marisaldi et al., 2014) and the Fermi space telescope (Briggs et al., 2010).

A careful analysis of BATSE, RHESSI and Fermi-GBM data permitted to identify some longer events, with durations longer than 1 ms (Smith et al., 2006; Dwyer et al., 2008; Cohen et al., 2010; Briggs et al., 2011). These events were not directly due to the detection of gamma-rays, but to secondary electrons and positrons produced by the TGF, and were called Terrestrial Electron Beams (TEBs). Contrary to gamma-rays, the charged particles are beamed by the magnetic field of the Earth, and can travel thousands of kilometers between one hemisphere to the other and may be detected in unusual locations for TGFs; e.g. the Fermi 091214 event detected above the Egyptian desert (Briggs et al., 2011; Sarria et al., 2016). These electrons/positrons can then be trapped by the geomagnetic field and they may provide a significant source of high-energy ($> 1$ MeV) particles to the radiation belts. The impact of TEBs on radiation belts still needs to be quantified.

In the near future, two missions are planned whose primary objective is the TGF detection : ASIM and TARANIS. The Atmosphere-Space Interaction Monitor (ASIM) is an European Space Agency (ESA) project with scientific leadership from the Technical University of Denmark (DTU) (Neubert et al., 2006). It will embark two X/Gamma Ray detectors (MXGS-LED and MXGS-HED) coupled with optical sensors (MMIA). It will be docked on the International Space Station (ISS) in the course of 2017. The Tool for the Analysis of RAdiation from lightNIng and Sprites (TARANIS) is a micro-satellite of the French Space Agency (CNES) that will be dedicated to the study of transient events related to thunderstorm activity (Lefeuvre et al., 2009) and will be launched in 2018. All instruments on-board the TARANIS spacecraft collaborate for the transient event study. Upon alert of one instrument, all instruments can record data prior, during and after the trigger. Two instruments have been specifically designed to study TGFs and TEBs : the instrument for X-Gamma-Ray and Relativistic Electrons (XGRE) and the Instrument for Detection of Energetic Electrons (IDEE). XGRE and IDEE are two of the four instruments that have the ability to trigger all the on-board instruments.

These two instruments will be presented in section 2. In section 3 we will make a comparison of the performances of XGRE and IDEE with those of RHESSI, Fermi-GBM and AGILE-MCAL, in the context of TGFs. Finally in section 4 we build a self-consistent picture to account for the detection rates of TGF and TEB seen by the satellites flying today, in order to estimate the future detection rates of TARANIS.

## 2 The TARANIS XGRE and IDEE instruments

### 2.1 The XGRE Instrument

XGRE can detect photons in the $[20 \, \text{keV} - 10 \, \text{MeV}]$ energy range and electrons in the $[1 \, \text{MeV} - 10 \, \text{MeV}]$ energy range. There are major differences when detecting photons (from TGF) and electrons or positrons (from TEB). Photons in this energy range always have a probability of not interacting with a given material, whereas an electron crossing a given material always deposits




**Figure 1. a.** Two side views of the TARANIS GEANT4 mass model. The three XGRE sensors are highlighted in red and the two IDEE detectors are highlighted in cyan. **b.** Exploded view of a XGRE sensor, showing the 4 detectors units (brown), the electronics and the aluminium housing. **c.** Cross section view of a XGRE detector unit, highlighting the sandwich design of plastic/LaBr$_3$ scintillators . **d.** Partial cross-section view of an IDEE detector's head, highlighting the position of the silicium and CdTe cells.

energy in it. A significant increase in photon energy always implies a significant increase in the average energy deposit on the detection material, allowing a proper estimate of the incident photon energy spectrum.

In general, the energy of an incident electron is difficult to estimate properly. Using several layers of detectors helps a lot but there remain uncertainties due to the detector's environment. Positrons will behave very similarly to electrons, with the addition that they will always annihilate into two 511 keV photons once they have lost most of their kinetic energy.




The XGRE instrument is presented in Figure 1. Figure 1.A. show its position on the TARANIS satellite (highlighted in red). XGRE is composed of three sensors, that are tilted by $20^o$ with regard to the payload baseplate. The relative counts of the three sensors allow to estimate the direction of the gamma-ray flux for bright events (more than 100 counts). Each sensor contains four Detection Units, as presented in Figure 1.b. . Each unit has one 8.7 mm thick lanthanum bromide crystal

($LaBr_3$) scintillator, surrounded by two 5 mm thick plastic scintillators, as shown in Figure 1.c., and the three scintillators are connected to two multi-anode photo-multipliers. This sandwich design allows the identification of the triggering photon or charged particle.

The plastic scintillators have a low effective atomic number ($Z \approx 6$) and a low density ($1.03 \mathrm{\,g/cm^3}$), therefore gamma-rays have a small probability to interact with it and/or to deposit all of their energy. On the other hand, gamma-rays have a much

higher probability of interacting with the $LaBr_3$ due to its high effective atomic number ($Z = 46.9$), its five times higher density ($5.08 \mathrm{\,g/cm^3}$) and its larger thickness. The three scintillators are sensitive to charged particles. If a significant amount of energy is deposited in the $LaBr_3$ only, it will probably be due to a gamma-ray. If some energy is deposited in a plastic only, it will likely be due to an electron with energy below 1.2 MeV. If energy is deposited in one (or two) plastic(s) and in a $LaBr_3$ crystal, it will likely be due to a higher energy electron (above 1.2 MeV).

The effective area of XGRE for detecting gamma-rays could be determined using our GEANT4 full mass model of the instrument and satellite. GEANT4 is a toolkit developped by a international collaboration lead by CERN to simulate the propagation of particles though matter (Agostinelli et al., 2003; Allison et al., 2006) . It is an essential tool to simulate high energy particle detectors and to estimate their performance.

Two side views of the GEANT4 mass model of XGRE are presented in Figure 1.a. This mass model will be refined in the

next years, using results of calibration campaigns.

To determine the response of the detector to X/gamma-rays, we drawn 150 mono-energetic beams of $20 \times 10^6$ photons, each with a different energy between 20 keV and 20 MeV. The particles are drawn from the direction of nadir, towards the satellite. Indeed, the attitude of the satellite is such that the detector will always point towards the Earth (nadir).

Any particle that deposits an energy above the electronic trigger is considered as detected, i.e. above 300 keV on a plastic

scintillator and/or above 20 keV in a $LaBr_3$ crystal.

Figure 2.a. shows the computed effective area of XGRE for gamma-rays, using $LaBr_3$ (black curve). The effective area of XGRE is maximal at $E_{max}$ is about 125 keV with an effective area of above 836 $cm^2$. Below $E_{max}$, the effective area decreases as weaker X-rays are more easily absorbed by materials surrounding the crystal (e.g. plastic scintillators, aluminium housing, hoods). The effective area is negligible below 20 keV by design. Above $E_{max}$, the effective area decreases and goes

down to $\approx 190$ $cm^2$ at 1.5 MeV. For higher energies, the pair production probability (by interaction with the detector or the surrounding material) becomes higher, increasing the effective area, that reaches $\approx 230$ $cm^2$ at 20 MeV.

In Appendix A, we describe how we can calculate average effective area, that is a unique value associated to a detector for detecting TGF or TEB. For the average effective area of XGRE, the calculation gives $\sigma_{XG}^{TGF} \approx 425$ $cm^2$. This value is indicated in Table 1, together with the values for the detectors of RHESSI, Fermi and AGILE, that will be discussed in section 3.





**Figure 2. a.** Effective area for X/gamma-rays versus energy, for the considered detectors. **b.** Effective area for electrons versus energy, for the considered detectors. **c.** Effective area for positrons versus energy, for the considered detectors.



To determine the response of the instrument to electrons (and positrons), we launch 150 mono-energetic beams of $20 \times 10^6$ electrons (or positrons), each beam with a different energy between 20 keV and 20 MeV. The electrons are drawn from two sides at $40^o$ from nadir and $40^o$ from zenith, that is representative of an average orientation of a magnetic field line seen by the satellite around equatorial regions (and this is also the orientation of the two IDEE detectors, see next section). The particles

are drawn around these two directions with an uniform randomization of $\pm 30^o$ for polar an azimuthal angles.

The simulation requires for an initial particle to make a deposit of at least 300 keV on a plastic scintillator to be detected, or at least 20 keV on a $LaBr_3$ scintillator. These deposits may be due directly to the electrons, or from bremsstrahlung secondary emissions they are producing.

The effective area of XGRE against electrons is shown by the blue curve of Figure 2.b. There is threshold energy $E_{XG}^t$ of

about 300 keV below which the area is very small (less than cm$^2$). Actually, each plastic scintillator is covered by a 0.6 mm thick hood made of Polyether ether ketone (PEEK). Electrons of 1 MeV kinetic energy will deposit about $E_{XG}^t$ in these hoods. About 10% of the area of the hood are covered by a 8 mm thick stiffener that will absorb more than $\sim 3$ MeV of kinetic energy from the electrons when they cross it. For energies higher than $E_{XG}^t$, the electrons can more likely reach the plastic scintillators because they will scatter to larger distances, and also bremsstrahlung emissions (that can be detected by the $LaBr_3$) become

more and more important, increasing the effective area from $\approx 100$ cm$^2$ at 600 keV to about 720 cm$^2$ at 20 MeV. Figure 2.c show the effective area of XGRE against positrons. It is essentially similar to the electron's curve, with addition of a constant value of about 280 cm$^2$. This constant value is due to positrons that annihilate (into two 511 keV photons) with the detector or some parts of the satellite.





To determine the effective area averaged over a TEB spectrum, $\sigma_{XG}^{TEB}$, we apply the method presented in Appendix A. The calculation gives about 233 cm$^2$ for XGRE. This value is presented in Table 1, together with the values obtained for the detectors of RHESSI, Fermi and AGILE, that will be discussed in section 3.

| | $\sigma^{TGF}$ (cm$^2$) | $\sigma^{TEB}$ (cm$^2$) |
|---|---|---|
| RHESSI total | 256[a] | 74 |
| AGILE-MCAL | 220[b] | 25 |
| Fermi-GBM BGO (1 unit) | 160[c] | 21 |
| Fermi-GBM NaI (1 unit) | 33 | 14 |
| Fermi-GBM total | 716 | 350 |
| XGRE | 425 | 233 |
| IDEE | 0 | 22 |
| TARANIS Total | 425 | 255 |

[a] Ref : Ostgaard et al. (2012)

[b] Ref : Marisaldi et al. (2015)

[c] Ref : Briggs et al. (2013)

**Table 1.** Summary of the TGF-spectrum averaged ($\sigma^{TGF}$) and TEB-spectrum averaged ($\sigma^{TEB}$) effective areas of RHESSI, AGILE, Fermi and TARANIS.

## 2.2  The IDEE instrument

The IDEE instrument is made of two electron detectors from 80 keV to 5 MeV energy. The main objectives of IDEE are to study Lightning-induced Electron Precipitations (Voss et al., 1984; Inan et al., 2007), and also the electrons beams associated to TGFs, as known as TEBs. The spectroscopy is possible up to 4.4 MeV, particles depositing more than this energy are counted in an overflow channel. As shown in Figure 1.d., each detector is made of 5 cells of silicium (Si) and 64 cells of cadmium telluride crystal (CdTe). The disposition of the cells into two layers allows to estimate the pitch angles of the incoming charged particles, by coincidence between Si and CdTe detectors, that will provide an important complementary information to the measurements of XGRE. One detector is pointed at $30^o$ from nadir, and the other at $40^o$ from zenith. The fields of view are $150^o \times 40^o$ for the Si part and $150^o \times 150^o$ pour for the CdTe part.

The Silicium cells have a geometrical area of 8 cm$^2$ (4 per detector), and are 0.3 mm thick. The CdTe cells are 5 mm thick and have a total geometrical area of 128 cm$^2$ for the sum of the two IDEE detectors, therefore most of the effective area will be due to the CdTe cells.

The effective area versus energy of IDEE for detecting electrons was estimated with the GEANT4 mass model, using the same methodology as for XGRE. An electron is detected if it deposits at least 80 keV on a Si cell or at least 350 keV on a CdTe cell. The effective area is presented in Figure 2.b. (black curve). It shows a threshold $E_I^t$ of about 610 keV below which the





effective area is essentially due to the Si cells, and therefore very small (less than 1 cm$^2$). $E_I^t$ is the sum of the 350 keV plus $\approx$ 260 keV, that is the average energy deposited on the 0.65 mm thick aluminium layer covering the CdTe cells. The effective area increases to about 28 cm$^2$ at 1.5 MeV, where all the electrons can cross the aluminium cover, and then goes up to about 20 cm$^2$ at 4 MeV and $\approx$ 38 cm$^2$ at 10 MeV, mostly due to the scattering of the particles to larger distance, and remains constant

for higher energies.

The effective area averaged over a TEB spectrum, $\sigma_I^{TEB}$, can be calculated to be about 22 cm$^2$ (see Appendix A for the method used). If IDEE or XGRE detects a TGF or a TEB event, they will be able to trigger the other instrument : we should consider the TARANIS spacecraft as a detector of TEB with an averaged effective area $\sigma_T^{TEB}$ of about 255 cm$^2$.

## 3   Comparison between instruments

In this section we present a comparison of the performances of XGRE and IDEE with those of RHESSI, Fermi-GBM and AGILE-MCAL, in the context of TGFs, without including CGRO-BATSE in the comparison. CGRO-BATSE detected 79 TGFs between the years 1991 and 2000, some of them being clearly identified as TEBs (Dwyer et al., 2008). But we did not include it because this number of TGF is significantly smaller than for RHESSI, Fermi-GBM and AGILE. Furthermore, BATSE only triggered on long events (it had a trigger window that could not be lower than 64 ms), over-estimated their durations and

under-estimated their brightnesses (Grefenstette et al., 2008; Gjesteland et al., 2010); thus making it a lot harder to separate between TGFs events and TEBs events, compared to the other instruments.

### 3.1   TGF detection performance

The Reuven Ramaty High Energy Solar Spectroscopic Imager (RHESSI) is a NASA spacecraft designed for the study of high energy radiation from the sun. It uses an array of nine high purity germanium detectors cooled down to nitrogen temperature.

A detailed description of the detector is presented in Lin et al. (2002); Smith et al. (2002). A response matrix of the RHESSI detector in the TGF context is publicly available from (see Section 7 on data availability). The provided matrix is already averaged for the spacecraft position and attitude. From this matrix, we can deduce the effective area versus energy of the detector, that is presented in Figure 2.a. As indicated in Ostgaard et al. (2012), RHESSI has an effective area for detecting TGF $\sigma_R^{TGF}$ of about 256 cm$^2$

The Astro-Rivelatore Gamma a Immagini Leggero (AGILE) is a satellite from the Italian Space Agency dedicated to the study of high energy gamma-ray (typically above 50 MeV) in the universe. The mini-calorimeter (MCAL) detector uses 30 cesium iodide (CsI) scintillator bars and can be used to detect lower energy gamma-rays (above 400 keV). It is presented into details in Tavani et al. (2009); Labanti et al. (2009). The effective area versus energy for AGILE-MCAL is taken from Marisaldi et al. (2015) and reproduced in Figure 2.a. As indicated in the same article, AGILE-MCAL has an effective area for detecting

TGF $\sigma_A^{TGF} = 220$ of about 220 cm$^2$.

The Gamma-Ray Burst Monitor (GBM) on-board the Fermi spacecraft is presented into details in Meegan et al. (2009). GBM is made of 12 sodium iodide (NaI) cylindrical detectors, sensitive in 20 keV - 10 MeV energy range, and 2 bismuth





germanate (BGO) cylindrical detectors sensitive from 200 keV to 40 MeV. Regarding the NaI detectors, the photons above 1 MeV are counted in a single channel and not used for spectroscopy, but they are included for the TGF counts and the search algorithm. The effective area for high energy photons can be calculated from an average of response matrices generated by the *gbmrspgen* tool, developed by the Fermi-GBM collaboration (see Section 7 Data availability) The response matrix of GBM for

a given event depends on the position and attitude of the spacecraft. To get an average effective area of the GBM detectors, we calculated an average matrix from 94 matrices that where generated from the 94 GBM triggered TGF of 2013. The effective area versus energy of the BGO and NaI detectors are presented in Figure 2.a.

As presented in Briggs et al. (2013), the effective area that should be used to detect TGF is 160 cm$^2$ for each BGO detector. Our calculation from the response matrices show that it should be about 33 cm$^2$ for each NaI detector, giving a total $\sigma_F^{TGF}$ of

about 716 cm$^2$ for Fermi-GBM. A summary of these averaged effective area for detecting TGF is presented in Table 1.

Below 30 keV, the NaI detectors of Fermi-GBM have the best effective area that ranges between 40 cm$^2$ and 300 cm$^2$. However, from simulation results it is not expected that TGF detected at satellite altitude show a lot of photons at these energies (Sarria et al., 2015), though this part of the spectrum has not been properly detected yet. Between 30 keV and 220 keV, XGRE has the best effective area (350-850 cm$^2$ ), that is about 1.4 higher than Fermi-NaI detectors, and 5 times higher

than RHESSI. For higher energies, it falls and goes below AGILE-MCAL and Fermi-BGOs at about 1 MeV ($\sim$280 cm$^2$) and below RHESSI at about 2 MeV ($\sim$250 cm$^2$). From around 760 keV, the effective area of AGILE-MCAL increases greatly, and reaches about 750 cm$^2$ at 20 MeV, making it about twice better than RHESSI and Fermi-BGOs and three times better than XGRE.

RHESSI, Fermi and AGILE suffered from issues related to the fact that their design is not perfectly suited to detect very

bright and short events such as TGFs. Concerning AGILE, this issue was likely solved after the disactivation of its anti-coincidence shield (Marisaldi et al., 2015). Depending of the processing algorithms and the electronics used by the detector, this can cause several issues, such as under-estimating the number of counts for bright TGFs (because of the detector's "dead time"), over-estimating the duration of bright TGFs, or incorrectly measuring photon energies (because of pulse "pile-up").

For Fermi GBM detectors, the nominal dead time lasts 2.6 $\mu$s, but it goes up to 10.4 $\mu$s if the overflow channel is filled,

i.e. there is a count with energy above 1 MeV on a NaI detector or a count with energy > 40 MeV on a BGO detector. NaI detectors have a high rate of overflow counts, making TGF spectra obtained from them very hard to analyse in practice. On the other hand, these problems are less important on the BGO detectors, allowing correction and study of spectra from single TGF events (Mailyan et al., 2016).

XGRE uses Lanthanum Bromide crystal scintillators coupled with fast electronics, resulting in a dead time of 350 ns and a

pile up time of 150 ns, giving a capability to count up to $\sim$ 9 photons/$\mu$s (each of the three sensors being independent), that should be enough to avoid dead time or pile-up issues up to a count rate of about 10 million counts per second. Thus XGRE should derive precise measurements of light-curves and spectra, even for the shortest TGF.

The dead time of IDEE is less than 4 $\mu$s. It should not suffer of important dead times issues when detecting TEBs, since they show about 20 times less particles/cm$^2$/ms at satellite's altitude compared to TGFs (see Figure 3), and IDEE also has a

relatively small effective area.


## 3.2 TEB detection performance

RHESSI, Fermi-GBM and AGILE-MCAL were not designed to detect electrons or positrons, therefore no response matrix is provided for these particles. Nevertheless, we could proceed to Monte-Carlo simulations of these detectors to get a basic idea of their performances for detecting TEBs. The RHESSI detectors are surrounded by several millimeters of aluminum (Dwyer

et al., 2012), that only very high energy electrons can cross.

Using a complete mass model of the RHESSI spacecraft (D. Smith, private communication), we could estimate its effective area for different electron incident energies. The procedure we followed is different that from XGRE/IDEE, since the orientation (attitude) of the spacecraft is not known and has no reason to point towards Earth like for TARANIS. Therefore we simply draw the particles randomly and uniformly over all directions around the spacecraft.

The effective area of RHESSI against electrons is displayed in Figure 2.b. It is $< 1$ cm$^2$ below 400 keV and then it rises to about $4$ cm$^2$ at 1 MeV and then increases with energy until it reaches about $500$ cm$^2$ at 20 MeV, due to important bresmstrahlung emissions. Using the same method as for XGRE, we could estimate an effective area $\sigma_R^{TEB}$ of RHESSI averaged for a typical TEB event of about $74$ cm$^2$.

Regarding Fermi-GBM, GEANT4 detailed models of single BGO and NaI detectors are available as GDML files as part of

the GRESS software (Kippen et al., 2007). A NaI detector is covered by an aluminium parts (including the PMT), and one side of the crystal has a 0.2 mm thick berylium window and a 0.7 mm thick silicone layer in-between the two. The BGO detector has some dense parts on both side (including the photo-multiplier tubes) and the rest is covered with a $\sim 3$ mm thick carbon fiber (CRFP), and maintained by two titanium rings.

These single detector models are not enough to estimate the reponse of Fermi-GBM to electrons, because they do not take

into account their accommodation on the spacecraft, nor the entire spacecraft (e.g. platform, subsystems, and LAT detector). We could not have access of to the full mass model of the Fermi satellite, but we could build a very simplified version, by looking to several Fermi-GBM documents; in particular (Meegan et al., 2009) and the references therein. Our simplified model contains the biggest parts of the spacecraft with approximative densities, and the 2 BGO and 12 NaI detectors are accurately placed. We think this model is reasonable for electrons since they get easily absorbed by the elements of the spacecraft, and

also we only need a basic estimation of the GBM response to electrons.

The response of GBM to mono-energetic electron beams is presented in Figure 2 b. We followed the same procedure as for RHESSI (the particles are drawn randomly and uniformly over all directions around the spacecraft). The effective areas show threshold energies ($E_{NaI}^t \approx 500$ keV, $E_{BGO}^t \approx 1.5$ MeV) below which the effective area is very small. Below these energies, the electrons or positrons can hardly reach the crystals, because they are absorbed by surrounding materials. Above these

threshold energies, the leptons have enough energy to have a chance to reach the crystals, and the effective area increases with increasing kinetic energy. This increase is because electrons with higher energy will scatter to higher distances in the spacecraft and will also produce more bremsstrahlung photons and with higher energies. For 20 MeV electrons, it reaches a value of about $770$ cm$^2$ for the sum of the 12 NaI and $325$ cm$^2$ for the sum of the two BGO. The response to positron is similar to the response





to electrons, with the addition of a constant value, that is about 690 cm$^2$ in this case. As for other instruments, we can use the method presented in Appendix A to calculate a TEB-averaged effective area for Fermi-GBM, that is $\sigma_F^{TEB} \approx 350\,\text{cm}^2$

Regarding AGILE, the full mass model was provided by the AGILE team (M. Marisaldi, private communication). The MCAL detector on the AGILE spacecraft is surrounded by several elements (e.g. the MITA spacecraft Bus, the GRID, the

Super-AGILE, the anticoincidence system or the carbon fiber structure surrounding the CsI bars) that will absorb a significant amount of energy of the electrons before they can reach the CsI crystals (Longo et al., 2002; Cocco et al., 2002; Labanti et al., 2009). We could perform simulation to check the response of MCAL to electron and positron beams, following the same procedure as for RHESSI and Fermi-GBM. The results are displayed in Figure 2.b and 2.c. (red curves). All the incident electrons with kinetic energies below about 3 MeV are absorbed before reaching the CsI bars. Above this energy, the effective

area increases with increasing energy, mainly due to the production of bremsstrahlung photons that can reach the detectors. It reaches $\sim 430$ cm$^2$ at 20 MeV, where a lot of bremsstrahlung photons are produced. As for the other instruments, the response to positron is similar to the response to electrons, with the addition of a constant value (about 100 cm$^2$ in this case) due to photons produced by positron annihilation with the spacecraft. It results in an effective area $\sigma_A^{TEB}$ averaged on a TEB spectrum of about 25 cm$^2$ for AGILE-MCAL. All the TGF and TEB effective areas are summarized in Table 1.

Concerning "dead times" or "pile-up" issues, all these detectors did not have any important problem concerning TEB. Indeed, the flux (particles/cm$^2$/ms) for a TEB event at satellite altitude is usually about 20 time less than for TGF (see Figure 3), and their averaged effective area are also several times smaller for electrons than for gamma-rays.

## 4 Estimating TGF/TEB detection rates

### 4.1 Past TGF and TEB detections

The AGILE TGFs of the second catalog are given between 03/23/2015 and 06/23/2015, and contains 279 TGFs (Marisaldi et al., 2015). Taking into account that TGF are slightly more likely to be detected during this time period than the average of the rest of the year, it corresponds to about $N_A = 1070$ TGFs/year. For RHESSI, the detection rate is about $N_R = 350$ TGF/year for the second catalog (Ostgaard et al., 2015). For Fermi, by looking to publicly available catalog (http://fermi.gsfc.nasa.gov/ssc/data/access/gbm/tgf/), we could estimate that about $N_F = 650$ TGFs/year were detected after

the offline searching method was set up (Briggs et al., 2013). All these values are summarized in Table 2.

Concerning Terrestrial Electrons Beams (TEBs), they were detected by RHESSI and Fermi. RHESSI detected clearly only two TEB events, and one of them was presented (Smith et al., 2006). This number is too low to permit an estimation of the number of TEB event that will be detected by TARANIS. As discussed in the previous section, Fermi-GBM has a much better sensitivity to electrons than RHESSI, and could detect about 24 events between August 2008 and February 2015, giving 3.7

TEBs/year.

No TEB event was reported by AGILE, and we speculate this is because the effective area for detecting TEB is not high enough ($\approx 25$ cm$^2$), and is actually mostly due to bremsstrahlung or annihilation photons produced by the electrons/positrons (see previous section).


## 4.2 Simulated Flux Profiles

Using the MC-PEPTITA Monte-Carlo model (Sarria et al., 2015), we estimated average flux profiles of gamma-rays and electrons detected by the satellites and associated to TGFs. The source is assumed to follow an energy spectrum $\propto 1/E \times exp(-E/(7.3\text{MeV}))$. The production altitude is uniformly sampled between 12 and 15 km and it is located at $(\theta = -13^o, \phi = 32^o)$ geodetic coordinates. The opening angle is uniformly sampled between $0^o$ and $40^o$. The source is also tilted by an angle $\psi$ that is uniformly sampled between 0 and $10^o$.

Concerning the time distribution of the source, there are currently two different results. On one hand, by comparing simulated TGFs with AGILE data, Marisaldi et al. (2015) suggests that, at the source, the TGF is created almost instantaneously, so that the TGF durations are mainly due to delays due to scatterings in the atmosphere, and long duration TGFs may be a succession of multiple pulses. On the other hand, by comparing simulated TGFs with Fermi data, Fitzpatrick et al. (2014) concluded that the source distribution is not created instantaneously for a vast majority of Fermi-GBM TGFs and indicated that a good fit to the Fermi data is a time distribution of the TGF source following a Gaussian (Normal) distribution with $\sigma = 50\,\mu$s. The results of MC-PEPTITA simulations suggest a source duration in-between an almost instantaneous source and a normal source duration of $\sigma = 50\,\mu$s . Assuming a normal distribution definition $\propto \exp\left(-t^2/\left(2\sigma^2\right)\right)$ for the TGF photons when they are produced, and using a standard deviation of $\sigma = 20\,\mu$s, results to $t_{90}$ durations of TGFs down to $\sim 60-70\,\mu$s; that corresponds to the lowest durations observed by the Fermi spacecraft (Fitzpatrick et al., 2014).

From Fermi data, Fitzpatrick et al. (2014) indicates that Fermi typically detects about 0.08 photons/cm$^2$ over a TGF duration of 200 $\mu$s (giving 0.4 photons/cm$^2$/ms) at a radial distance of 500 km between the satellite position and the position of the source of the TGF. We used this value to give a scale to the flux distributions presented in Figure 3, that is obtained by assuming that $4.4 \times 10^{17}$ photons (with energies > 20 keV) are produced at the source.

The flux profiles resulting from the simulations are presented in Figure 3.a. The fluxes are presented as a function of the radial distance ($r_d$) between the source position of the TGF (projected at the altitude of the satellite) and the satellite. The three presented altitudes approximately correspond to AGILE (490 km), Fermi (550 km) and TARANIS (700 km). The fluxes are expressed in terms of particles/cm$^2$/ms, considering photons and electrons. Figure 3.b. presents the corresponding time durations ($t_{90}$).

Below $r_d =$100 km, the photon flux at an altitude of 490 km is about 31 % higher than the flux at 560 km, and twice the flux at 700 km. This difference of fluxes corresponds to the $1/R^2$ variation expected from an isotropic point source detected from various distances. At about $r_d =$300 km, the photon fluxes are similar for the three altitudes. Above $r_d \approx 600$ km, the flux at 700 km is about 57% higher than the flux at 560 km, and the flux at 560 km is 39% times higher than at 490 km altitude.

Concerning electrons, the fluxes are close at the three considered altitudes, so we only represented the flux at 550 km. It is important to note that the time scattering of electrons detected at satellite altitude can vary significantly depending on the length of the magnetic field line the particles have to travel, that depends on the coordinates of the spacecraft (higher absolute latitudes usually meaning longer field lines). This time dispersion is because electrons are produced with various pitch angles and energies, that will imply various propagation speeds along the geomagnetic field lines (Dwyer et al., 2008; Sarria et al.,





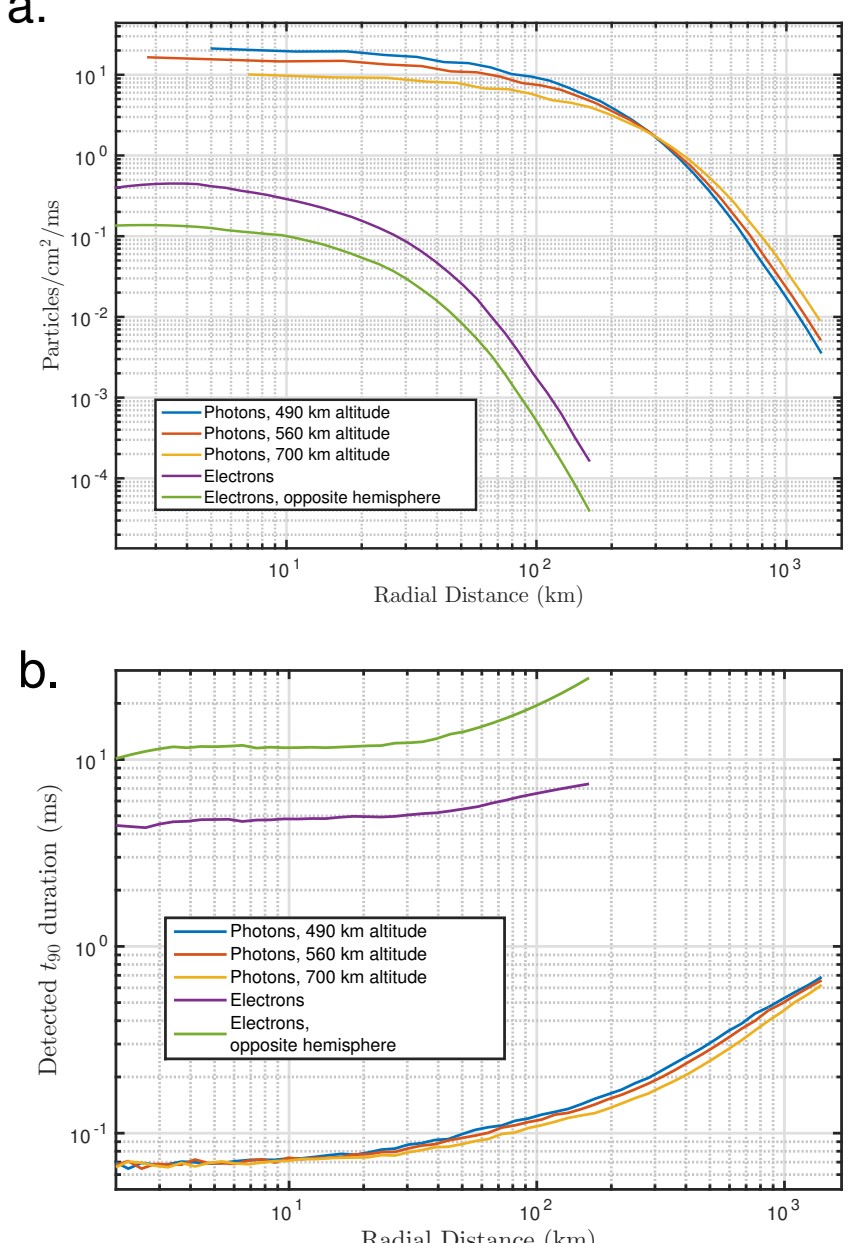

**Figure 3.** Results of MC-PEPTITA simulations. **a.** Flux profile versus radial distance between the TGF source (projection at satellite altitude) and the satellite; for photons (various altitudes) and electrons (seen at the hemisphere of production and in the opposed hemisphere). The two electron profiles correspond to a detection altitude of 560 km, but are very close for 490 or 700 km altitude. **b.** $t_{90}$ durations associated to the flux profiles. Like for the flux profiles, the electron distributions stay very similar at 490 or 700 km altitude.





2016). The results shown here are for a given magnetic field line of about 6000 km length, that can be roughly considered as an average TEB case. In Figure 3, the flux of electrons is about 3 times higher if detected in the hemisphere where the the TGF is produced. The spatial fluxes (electrons/cm$^2$) are actually quite close on both hemispheres, but, as showed in Figure 3.b., the $t_{90}$ time durations of the TEBs are about 3 times higher in the opposite hemisphere.

**4.3 Estimating a map of TGFs that can be detected by satellites.**

We propose to build an approximative map of TGFs that can be detected by satellites, based on the TRMM-LISS and OTD global lightning density map (Cecil et al., 2014). Compared to this distribution, it was noticed that the TGF density detected by satellites tends to be higher towards the equator. Actually, this is supposed to be due to the fact that the tropopause is higher for latitudes closer to zero, where TGF photons have to cross less atmosphere before reaching space, and can be more easily

detected by satellites. Let $\rho_L(\theta, \phi)$ be the lightning density from the LISS/OTD database for a given latitude and longitude. Let $T(\theta)$ be an approximative profile of the tropopause height as function of latitude. We used data obtained from Lewis (2009) that was fit by a simple Normal distribution $T(\theta) \propto \exp\left[((\theta - \theta_0)/2\sigma_\theta)^2\right]$ with parameters $\theta_0 = 0.8746^o$ and $\sigma_\theta = 37.49^o$. Then the TGF density at a given latitude and longitude is assumed to follow :

$$\rho_{TGF}(\theta, \phi) \propto \rho_L(\theta, \phi) \times [T(\theta)]^\beta \qquad (1)$$

The proportionality sign ($\propto$) denotes the fact that we do not define an absolute scale for this density, and therefore all the estimations given afterwards will only use ratios of summed values of $\rho_{TGF}(\theta, \phi)$. Equation (1) shows that the tropopause profile is set at a power $\beta$ that we are using as a free parameter, that will be adjusted to get the best possible agreement between this simple model and the observations. This estimated TGF global production map is displayed in Figure 4. This Figure also shows the groundtracks of RHESSI, Fermi, AGILE and TARANIS (planned). They have been calculated using the

Two-Line Element from the CELESTRACK database (https://www.celestrak.com/NORAD/elements/). The orbit of TARANIS is assumed to be similar to that of the DEMETER satellite. RHESSI, Fermi and AGILE show equatorial orbits with various inclinations ($38^o$, $25.6^o$ and $2.5^o$ respectively), whereas TARANIS will follow a quasi-polar sun-synchronous orbit. This orbit is not the best for detecting TGFs since it will have a significantly reduced coverage of the equatorial region where TGF are more likely to be detected, and the impacts on TGF detection rates are discussed in the next section. However, the orbit of

TARANIS covers almost uniformly all latitudes and should permit to determine a global distribution of TGF without orbital bias.

**4.4 Estimating TGF and TEB Detection Rates**

Each detector will have a minimal threshold of counts $n_X^{min}$ for any detected event to be considered as a "detected TGF". Gjesteland et al. (2012) indicates that $n_R^{min} = 11$ (for the RHESSI second catalog) and Ostgaard et al. (2012) indicates $n_F^{min} =$

19 for Fermi-GBM. The value for AGILE ($n_A^{min} = 10$) is found by the TGF that has the lowest number of count in the AGILE second catalog (Marisaldi et al., 2015). The value of $n_G^{min}$ for TARANIS-XGRE is assumed to be also 10 but is hard to predict





**Figure 4.** Estimated global detectable TGF density map, and groundtracks of the orbits of TARANIS (planned), RHESSI, Fermi, and AGILE. The dark area denotes an approximative South-Atlantic Anomaly assumed for the simulations, and where TGF can occur but no detection by satellites is possible due to high background.

and will depend on the in-flight background. We should wait for the instrument to be launched to be able to know precisely which value will be used.

The ratio between this threshold value $n_X^{min}$ and the averaged effective area $\sigma_X^{TGF}$ gives a limit of sensitivity for the instrument. Combining this limit with the radial distance flux profiles (section 4.2 and Figure 3), we can deduce a limit distance 5 $R_X^{lim}$ corresponding to this sensitivity. The limit of sensitivity of Fermi is $\approx 0.053$ photons/cm$^2$/ms ( = 19 photons / 716 cm$^2$ / 500 $\mu s$) and corresponds to a limit radius of $R_F^{lim} \approx 795$ km. This value is consistent with the maximum distance between the TGF source positions and Fermi footprints given using WWLLN associations (Briggs et al., 2013; Fitzpatrick et al., 2014). We can also determine $R_R^{lim} = 694$ km that is reasonably close to the largest distance found between RHESSI's position and the WWLLN match of the TGF source location (Nisi et al., 2014). Using the simulated photon flux and time profiles at 490 km





altitude, we could estimate $R_{XG}^{lim} = 648$ km for AGILE-MCAL. Using the simulated photon flux and time profiles at 700 km altitude, we could estimate $R_{XG}^{lim} = 917$ km for TARANIS-XGRE (corresponding to a $t_{90}$ of about 420 $\mu$s). All theses values are summarized in Table 2.

Knowing $R_X^{lim}$, the orbit of the satellite, and the detectable TGF density map, we can deduce a detection efficiency, expressed as $E_{X/A}$. Since we did not find an absolute scale for it, we only expressed it relatively to AGILE (second catalog), which has the highest TGF detection rate.

This detection efficiency is computed using the following algorithm :

– We consider a step of time $\delta_t =120$ seconds, that is small enough compared to the scale of duration of one orbit of about 5400 seconds.

– At each time step corresponds a position of the satellite $(\theta_i^X, \phi_i^X)$.

– at each position, the TGF densities from the map $(\rho_{TGF}(\theta, \phi))$ are summed within a radius of $R_{lim}^X$ around $(\theta_i^X, \phi_i^X)$, giving a quantity $\Sigma^X$.

– $\Sigma^X$ is incremented this way over 48 hours (1442 steps).

– if $(\theta_i, \phi_i)$ is inside the South-Atlantic Anomaly (SAA), $\Sigma^X$ is not incremented. We use the approximative SAA area presented in dark in Figure 4.

The ratio $\Sigma^X/\Sigma^A$ (between a given satellite X and AGILE) gives a detection efficiency $E_{X/A}$, whose values are summarized in Table 2. Applying these efficiencies to the AGILE detection rate gives detection rate estimates of 649.2 TGFs/year for Fermi and 349.7 TGFs/year for RHESSI. For both, the relative difference with the observed detection rates are less than 1%. Note that the value of the parameter $\beta$ for the detectable TGF map (see section 4.3) was adjusted to $\beta = 7.1$ in order to minimize these differences.

One last parameter to be taken into account is the diurnal cycle of lightning. Lightning activity was found to be non-uniform with local time and has maximum around 17h and minimum around 11h (Cecil et al., 2014). TARANIS, with its sun-synchronous orbit will always be at a local time between 22h30-2h and 10h30-14h. The other satellites have equatorial orbits therefore their local time is almost uniformly distributed between 0h and 24h. It implies that XGRE will miss an extra



24% of TGF compared to the other satellites. Finally, our estimation gives about 225 TGFs/years for TARANIS. All the important parameters used for this estimation are summarized in Table 2.

| | $h_X$ (km) | $n_X^{min}$ (counts) | $R_X^{lim}$ (km) | $E_{X/A}$ | $N_X^{TGF,obs}$ (TGFs/year) | $N_X^{TGF,est}$ (TGFs/year) |
|---|---|---|---|---|---|---|
| RHESSI (Second catalog) | 565 | 11 | 694 | 32.7 % | 350 | 349.7 |
| Fermi-GBM | 543 | 19 | 795 | 60.7 % | 650 | 649.2 |
| AGILE-MCAL (Second catalog) | 491 | 10 | 648 | 100 % | 1070 | 1070 |
| TARANIS-XGRE | 700 | 10 ? | 917 | 27.64 % | ? | 225[a] |

[a] Takes into account the diurnal correction.

**Table 2.** Altitudes, detection count thresholds, limit radii, detection efficiencies and number of TGFs per year (observed and estimated) for the considered satellites.

The catalog of Fermi GBM TEBs presents 24 events between 08/07/2008 and 02/02/2015, giving $N_F = 3.7$ TEBs/year. These events present a minimum count of $n_F^{min} \approx 150$ for the event ID TEB130521580. Since XGRE will discriminate electrons from photons, this threshold should be similar to TGF, i.e. $n_{XGRE}^{min} = 10$. As in the case of TGFs, this value is hard to estimate and the correct value will only be known after in-flight tests of the instrument. As discussed in section 4.2, the effective area of TARANIS (XGRE+IDEE) for detecting TEB could be estimated : $\sigma_G^{TEB} = 255, cm^2$. This TEB average effective area could be estimated for Fermi-GBM, giving $\sigma_R^{TEB} \approx 350\,cm^2$.

To determine the TEB detection efficiency, the algorithm presented for the TGF case has to be modified. If the satellite is located at given coordinates, the considered density is not the density at this point, but the sum of the two densities located at the two magnetic footprints of the field line. These coordinates are determined from MC-PEPTITA runs that can track the electrons in the geomagnetic field. In these simulations, the electrons are drawn at 100 km altitude with various pitch angles (this altitude being approximately the altitude where the secondary electrons from TGF can escape Earth's atmosphere).





From all this information, we can calculate detection efficiencies : between TARANIS and Fermi $E_{T/F}^{TEB} = 885\%$. As for the TGF estimation, we also have to account for the diurnal effect for the TARANIS case, and our final estimate is about 25 TEBs/year. All the important parameters used for this estimation are summarized in Table 3.

| | $n_X^{min}$ (counts) | $R_X^{lim}$ (km) | $E_{X/F}$ | $N_X^{TEB,obs}$ (TEBs/year) | $N_X^{TEB,est}$ (TEBs/year) |
|---|---|---|---|---|---|
| Fermi-GBM | $\sim 150$ | 23 | 100 % | 3.7 | 3.7 |
| TARANIS (XGRE+IDEE) | 10 ? | 72 | 885 % | ? | 25[a] |

[a] Takes into account the diurnal correction.

**Table 3.** Count thresholds, limit radii, detection efficiencies and number of TEBs per year (observed and estimated) for the considered satellites.

## 5 Conclusions

The TARANIS spacecraft will have two important instruments to study TGFs and TEBs : XGRE and IDEE. XGRE will detect both electrons and X/gamma-rays, with the ability to discriminate one type of particle from the other. The IDEE instrument is focused on electrons, with the ability to estimate their pitch angle. Both instrument will be able to trigger one another.

Using Monte-Carlo simulations, mass models and a standard TGF spectrum, we could estimate that XGRE will have about 425 cm$^2$ effective area for detecting TGFs. The combination of XGRE and IDEE will give about 255 cm$^2$ effective area for detecting electrons associated to TGFs. With a count rate capability of about 10 million counts/second, XGRE should suffer of much less "dead time" issues during bright TGF events, that were detrimental for previous detectors. Thus XGRE should derive precise measurements of light-curves and spectra, even for the shortest TGF.

Using Monte-Carlo simulations of the TARANIS, RHESSI AGILE, and Fermi spacecrafts, we could estimate the response of their detectors to electrons and positrons, and provide a quantitative comparison between them. By combining this knowledge with an approximative world map of detectable TGF density and with MC-PEPTITA Monte-Carlo simulations of TGF propagation in the atmosphere, we could build an accurate model of the TGF detection rates of RHESSI, AGILE, and Fermi. It could be used to estimate that TARANIS should detect about 225 TGFs/year and 25 TEBs/year.

## 6 Code availability

The GEANT4 mass model of TARANIS satellite with XGRE and IDEE instrument is still under developpement and is not publicly available yet. But simulations in specific configurations can be requested to the corresponding author, contact David Sarria (dsarria@apc.in2p3.fr).

The GEANT3 mass model of the RHESSI detector and spacecraft can be requested to David Smith (dsmith@scipp.ucsc.edu)



The GEANT3 mass model of the AGILE detectors and spacecraft can be requested to Martino Marsaldi (Martino.Marisaldi@uib.no), Marcello Galli (marcello.galli@enea.it) and Francesco Longo (franzlongo1969@gmail.com).

MC-PEPTITA simulations can be requested, contact David Sarria (dsarria@apc.in2p3.fr). MC-PEPTITA program was developed under a contract of Centre National d'Etudes Spatiales (CNES) and Direction Generale de l'Armement (DGA), whose

permissions are required in order to get access to the source code.

## 7   Data availability

The data generated for this work can be requested to the corresponding author, contact David Sarria (dsarria@apc.in2p3.fr).

The $V2.3.2014$ gridded satellite lightning data were produced by the NASA LIS/OTD Science Team (Principal Investigator, Dr. Hugh J. Christian, NASA / Marshall Space Flight Center) and are available from the Global Hydrology Resource Center

(http://ghrc.msfc.nasa.gov).

The response matrices of Fermi GBM detectors are publicly available using the *gbmrspgen* tool, whose utilisation is detailed in the following website https://fermi.gsfc.nasa.gov/ssc/data/analysis/scitools/gbmrspgen.html

The response matrix of the RHESSI detector is publicly available in the following website http://scipp.ucsc.edu/~dsmith/tgflib/public/.

## 15   Appendix A:  Determining detectors' averaged effective areas

We use a custom method to determine the average effective area of an instrument for detecting TGFs (or TEBs). Using a simulation of a given instrument, we can launch mono-energetic beams of particles of energy $E$ and determine the number that has been detected $N_d(E)$. We can then determine $S_X^Y(E)$, the effective area at the energy $E$, where $Y$ corresponds to the event type (TGF or TEB) and $X$ designates the detector (XGRE, IDEE, RHESSI, GBM or MCAL). Assuming there are $N_{launch}$

particles drawn uniformly from an area $S_{launch}$ (that should be higher than the area of the whole satellite) :

$$S_X^Y(E) = N_d(E) \times \frac{S_{launch}}{N_{launch}} \tag{A1}$$

$S_X^Y(E)$ can then be averaged over an assumed spectra of TGF (or TEB) to obtain a value $\sigma_X^Y$, characterizing the average effective area of the detector for detecting a TGF (or TEB) :

$$\sigma_X^Y = \frac{\int_{20\,kev}^{20\,MeV} f_Y(E)\, S_X(E)\, dE}{\int_{20\,kev}^{20\,MeV} f_Y(E)\, dE} \tag{A2}$$

Where $f_Y(E)$ is the assumed spectrum of the considered event type. We choose to use the photon and electron spectra at satellite altitude presented in the Figure 4 of Dwyer et al. (2008), assuming it is valid for all the orbits of previously mentioned experiments. We also assume that 10% of the electrons are actually positrons (as it is estimated from simulations of Terrestrial



Electron Beams (Sarria et al., 2015)), and that the electron spectrum does not differ very much in its shape from the positron spectrum.

*Author contributions.* David Sarria prepared most of the manuscript. David Sarria, Remi Chipaux, Jean-Pierre Baronick contributed to the GEANT4 model of the XGRE instruments and TARANIS satellite. David Sarria performed all GEANT4 simulations (TARANIS

XGRE, IDEE with satellite, and Fermi-GBM). Francois Lebrun, Pierre-Louis Blelly and Philippe Laurent provided a detailed review of the manuscript, and important feedback on the XGRE instrument, and instrument comparison. Remi Chipaux provided a review of the manuscript. Damien Pailot and Miles Lindsey-Clark provided important data to validate the GEANT4 model of the XGRE instrument, as well as important feedback on the instrument description (Section 2.1). Jean-Andre Sauvaud and Pierre Devoto provided the GEANT4 model of IDEE, important data to validate it, and contributed to the IDEE instrument part of the manuscript (Section 2.2). Philippe Laurent per-

formed GEANT3 simulation on the RHESSI and AGILE mass models, that where provided by David Smith (RHESSI); Martino Marsaldi, Marcello Galli and Francesco Longo (AGILE).

*Acknowledgements.* We thanks D. Smith for providing the mass model the RHESSI Spacecraft, and discussion about its response to electrons. We thank M. Briggs for his help in estimating of the Fermi-GBM reponse to electrons. We thank M. Marisaldi, F. Longo and M. Galli for providing the mass model of the AGILE Spacecraft, and their help in discussing the response of MCAL to electrons.

This work was granted access to the HPC resources of CALMIP supercomputing center under the allocation 2015-p1505.

We thank the CNES (Centre National d'Etudes Spatiales) for its financial support.



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
