# Peer review of "TARANIS XGRE and IDEE Detection Capability of Terrestrial Gamma-Ray Flashes and Associated Electron Beams"

_Geoscientific Instrumentation, Methods and Data Systems, 2017_

## Referee Comment (RC1) · Anonymous Referee #1 · 14 Apr 2017

In this paper, the authors report an evaluation of detector efficiencies, from which they infer Terrestrial Gamma ray Flashes (TGFs) and Terrestrial Electron Beams (TEBs) detection rates for the current space missions RHESSI, AGILE, and Fermi, and for the planned mission TARANIS. The paper is timely, well-written, presents a rigorous work, and is logically articulated. It is of high interest to the High-Energy Atmospheric Physics community. Figures are of good quality and the paper is fairly referenced. For these reasons, I strongly support it for publication in *Geoscientific Instrumentation, Methods and Data Systems*. Minor revisions are suggested below.

[Figure]

Main comments:

1. The detection constraint is defined by the number of photons detected by the instrument. It is understandable that comparing the efficiency of various detectors of various kinds and associated with different electronics is not an easy task. However, it would be important to comment on the role of the duration of the "discovery bin" to be used to accumulate $n_X^{min}$ counts, and the related comparison between different detectors. What is the impact of the discovery bin on the results? It seems that addressing this question is possible using the results shown in Figure 3b.

2. p. 4, ll. 21–23. What would be the impact of a distribution of tilted broad-beam sources (such as that assumed in Section 4.2).

3. In the whole text. It is important to give readers an idea about the accuracy of the effective areas and TGF detection rates estimated. Please indicate uncertainties on these results anytime they are given in the text, in the tables, and if they are sufficiently large, in the figures as error bars.

Line-by-line comments:

1. p. 4, l. 1. "Figure 1.A." to be changed in "Figure 1.a.". Please verify the typography in the whole paper.

2. p. 4, l. 16. Please replace "by a international collaboration lead by CERN" with "by an international collaboration led by CERN". Please verify grammar typos in the whole paper.

3. Caption of Figure 2.a. Please indicate all the references or sources for the various results presented in this figure.

4. p. 6., l. 3. The viewing angles mentioned here are not consistent with those p. 7, l. 12.

5. p. 6., l. 9. "blue curve" needs to be changed with "black curve".

6. p. 7., l. 19. "black curve" needs to be changed with "blue curve". Please check consistently in the whole paper.

7. p. 8., l. 19. "cooled down to nitrogen temperature" likely needs to be changed with "cooled down to liquid nitrogen temperature".

8. p. 8., l. 21. Please complete "publicly available from ..."

9. p. 10, l. 6. Please indicate the year of the private communication.

10. p. 11., l. 3. Please indicate the year of the private communication.

11. p. 12, l. 3. Please provide references for justifying your choices for the source spectrum, the source altitudes, the geodetic coordinates, the opening angle of the beam, an the tilt angle of the source.

12. p. 12, l. 15. Please indicate that the $t_{90}$ durations obtained here are consistent with Compton scattering in the atmosphere acting on a source lasting $\sim$20 $\mu$s [Celestin and Pasko, Geophys. Res. Lett., 39, L02802, 2012].

13. p. 14, l. 8. "supposed" probably needs to be changed with "assumed".

14. p. 14, ll. 7–10. There are a few references to cite at this point about the effects of the tropopause heights [e.g., Smith et al., J. Geophys. Res., 115, A00E49, 2010; Nisi et al., J. Geophys. Res., 119, 8698–8704, 2014].

15. p. 15, Figure 4. It would be convenient if landmasses were drawn on this map.

16. p. 16, l. 1. "$R_{XG}^{lim}$" should probably be replaced with "$R_A^{lim}$". Please check consistently in the whole paper.

17. p. 16, l. 11. "$R_{lim}^{X}$" subscripts and superscripts are inverted as compared to the notation used in the rest of the text. Please check consistently in the whole paper.

---

## Referee Comment (RC2) · Anonymous Referee #2 · 21 Apr 2017

This paper presents Monte Carlo calculations of detection rates for Terrestrial Gamma ray Flashes (TGFs) and Terrestrial Electron Beams (TEBs) for a number of current space missions (RHESSI, AGILE and Fermi) and for the upcoming TARANIS mission. The paper demonstrates convincingly that TARANIS can be expected to contribute significantly to the field, especially with respect to the expected number of detected TEBs. The paper, which is well-written is therefore of high interest to the High-Energy Atmospheric Physics community and I can recommend that it for publication in Geoscientific Instrumentation, Methods and Data Systems.

Below I have listed suggested improvements and clarifications

1. The quality of Fig 1 makes it is very difficult to visualize especially the IDEE instrument. The figure ought to be improved. Define also the meaning of a Si cell and a CdTe cell.

2. The XGRE sensors are tilted by 20° to the base plate; but it is not clear how they are placed. Discuss also the expected angular resolution on the TGF direction determination using this arrangement.

3. The XGRE is planned for photons up to 10 MeV, however, the discussion and plots operates with energies up to 20 MeV, for IDEE the discrepancy is even greater since the max energy is 5 MeV. I suggest that at least the plots indicate the energy range of the instruments, e.g. using dashed/dotted lines outside energy range.

4. It is not clear why the total effective area of the IDEE Si detectors is an order of magnitude smaller than their total geometrical area at 610 keV as explained p.8, l1 and shown in Fig 2b. 600 keV electrons will deposit much more than 80 keV in the Si detectors and therefore they will be detected. Comments and clarification are required.

5. It is unfortunate that the authors did not try to estimate the in-flight background. For the Taranis quasi-polar orbit it is expected that the background rates will vary significantly also outside the SAA. Therefore the threshold of counts value nmin will vary as well and a single value (e.g. 10 as assumed) might be insufficient. A discussion of the TGF trigger algorithm with respect to a varying background would be useful.

6. For clarity I strongly suggest to include outlines of the continents in Fig. 4. The SAA is here drawn in grey color and appear not dark as claimed in the caption. The same is the case for the Taranis orbit which is grey and not black as indicated in the legend.

I support the line-byline comments given by Referee #1 with a few additional ones:

1. p. 4, l. 21 we drawn -> we have drawn

2. p.4, l. 27 Emax is about 125 keV -> Emax ( ∼125 keV)

3. p.7, l.14 have a geometrical area of 8 cm2 (4 per detector) -> have a total geometrical

area of 8 cm2 (4 cm2 per detector)

---

## Author Comment (AC1) · 11 Jun 2017

We thank the Referee for his careful reading, and the valuable comments and suggestions he/she made to improve the quality of the manuscript. We considered each of the comments and provided an adequate answer.

The archive "gi-2017-1-supplement.zip" contains three files :

- answer_referee1.pdf : comprehensive answser to the referee 1 comments

- xgre_idee_sarria_revised.pdf : the revised article

[Figure]

- xgre_idee_sarria_latex_diff.pdf : manuscript highlighting the differences between the former and new version of the manuscript

Please also note the supplement to this comment:
http://www.geosci-instrum-method-data-syst-discuss.net/gi-2017-1/gi-2017-1-AC1-supplement.zip
* * *

---

## Author Comment (AC2) · 11 Jun 2017

We thank the Referee for his careful reading, and the valuable comments and suggestions he/she made to improve the quality of the manuscript. We considered each of the comments and provided an adequate answer.

The archive "gi-2017-1-AC2-supplement.zip" contains a folder with three files :

- answer_referee1.pdf : comprehensive answser to the referee 2 comments

- xgre_idee_sarria_revised.pdf : the revised article

[Figure]

- xgre_idee_sarria_latex_diff.pdf : manuscript highlighting the differences between the former and new version of the manuscript

Please also note the supplement to this comment:
http://www.geosci-instrum-method-data-syst-discuss.net/gi-2017-1/gi-2017-1-AC2-supplement.zip